# Recurrent *Scedosporium apiospermum* Cutaneous Infection in a Patient with Rheumatoid Arthritis: The Potent Role of IL-6 Signaling Pathway Blockade: A Case-Based Review

**DOI:** 10.3390/jof9060683

**Published:** 2023-06-18

**Authors:** Antigone Pieta, Aliki I. Venetsanopoulou, Christos Kittas, Eirini Christaki, Paraskevi V. Voulgari

**Affiliations:** 1Department of Rheumatology, School of Health Sciences, Faculty of Medicine, University of Ioannina, 45110 Ioannina, Greece; 2Microbiology Laboratory, School of Health Sciences, Faculty of Medicine, University of Ioannina, 45110 Ioannina, Greece; 31st Division of Internal Medicine & Infectious Diseases Unit, University General Hospital of Ioannina, Faculty of Medicine, University of Ioannina, 45110 Ioannina, Greece

**Keywords:** fungal infection, *Scedosporium apiospermum*, rheumatoid arthritis, IL-6 inhibitor, immunosuppression

## Abstract

Rheumatoid arthritis (RA) patients deal with a higher risk of bacterial and fungal infections compared to the general population because of their dysregulated immune system as well as the immunosuppressive therapy they usually receive. *Scedosporium* spp. is a fungal pathogen responsible for cutaneous, lung, central nervous system, and eye infections, mostly in immunocompromised patients, leading to death in disseminated cases. We report the case of an 81-year-old woman with rheumatoid arthritis treated with steroids and an IL-6 inhibitor who was diagnosed with scedosporiosis of the upper limb. She was treated with voriconazole for one month, which was discontinued due to adverse events, and when scedosporiosis relapsed, she switched to itraconazole. We also reviewed the current literature on RA patients presenting with *Scedosporium* infections. Early and accurate diagnosis of scedosporiosis has therapeutic and prognostic implications, as traditionally this fungus is resistant to commonly used antifungals. Clinical alertness regarding uncommon infections, including fungal, in patients with autoimmune diseases on immunomodulatory agents is essential for effective treatment.

## 1. Introduction

Rheumatoid arthritis (RA) is a chronic, autoimmune, and inflammatory disease, mainly affecting the joints of the body but also presenting a variety of extra-articular manifestations. About 0.5–1% of the population is affected, with a women-to-men ratio of 3:1 [1,2,3].

Many inflammatory pathways have been implicated in RA pathogenesis [1]. Some of them have been blocked successfully with the use of biological agents, resulting in low disease activity or remission. The cost that rheumatologists are indebted to is the higher risk for bacterial and fungal infections compared to the general population [4]. Studies regarding safety surveillance of biologic therapies, such as TNF-a inhibitors, biologic drugs targeting proinflammatory cytokines [interleukin (IL)-1, IL-6], anti-B-cell depletion therapies, and Janus kinase inhibitors (JAKis), showed, especially for fungal infections, that the hazard may be limited, if not underestimated. Still, the risk of fungal infection is present, and in disseminated cases, the outcome could be fatal [5]. As a result, in patients with dysregulated immune systems, the addition of a biologic drug suppresses the overreacting immune system, further impairing their ability to effectively fight against pathogens. A defective immune response could predispose to devastating infections or overwhelming inflammation through cytokine overproduction. It is common practice that these patients use a combination of immunomodulatory agents, including conventional disease-modifying antirheumatic drugs (DMARDs) and systemic steroids, which impose an added risk for fungal infections [6]. On the other hand, the hypothesis of an inflammatory syndrome accompanying fungal infections, especially after the acute phase, led some investigators to the hypothesis that blocking the IL-6 signaling pathway, among others, could interrupt an exaggerated immune response, inducing favorable outcomes [7].

Among fungal infections, the genus *Scedosporium* has gained visibility over the last three decades worldwide due to its recognition as a cause of infection both in immunocompetent and immunocompromised individuals, leaving behind the reputation of the opportunist [8]. Though the exact epidemiology is still unknown, the first to describe a fast-growing hyphomycete isolated from a white-grain mycetoma was Tarozzi in 1909, to be followed by Saccardo in 1911, who suggested the name *Scedosporium* for this newly recognized fungus [9,10]. Nowadays, six species affect humans: *Scedosporium apiospermum*, *Pseudallescheria boydii* (*Scedosporium boydii*), *Scedosporium aurantiacum, Scedosporium minutisporum*, *Scedosporium dehoogii*, and *Lomentospora prolificans*, which can be isolated from damp soil, decaying vegetation, polluted waters, sewage, and the manure of farm animals [10,11]. These species are acquired through inhalation or traumatic inoculation (wounds, scratching lesions, local injections or catheter fixation, surgical interventions) and may cause either localized (arthritis, osteomyelitis, sinusitis, keratitis, endocarditis, pneumonia, central nervous system, cutaneous and lymphocutaneous signs and symptoms) or systemic manifestations [12,13,14,15]. The most frequent types of *Scedosporium* infections are cutaneous and pulmonary. In immunocompromised patients, it is commonly found in those with a history of bone marrow or solid organ transplantation, suffering from cystic fibrosis, or receiving corticosteroids, engraving a subacute to chronic course. In immunocompetent individuals with pulmonary scedosporiosis, it has been reported in pre-existing tuberculous cavities or bronchiectasis following an acute course [16]. Allergic bronchopulmonary disease, linked to *Scedosporium* colonization in cystic fibrosis patients, has also been described in the literature [17,18]. Regarding its virulence, enhanced conidiation by contact with human cells, as well as structural components of conidia and hyphae such as peptidorhamnomannan and proteolytic enzymes, play a pivotal role in colonization and tissue invasion [19]. The isolation and identification of *Scedosporium* species is vital, as they display intrinsic resistance to traditional antifungals such as liposomal amphotericin B, and prompt initiation of appropriate treatment is crucial [20]. The diagnosis could be challenging due to the great similarity of clinical and histopathological features among *Scedosporium* spp. and other fungi [21,22]. Requisite characteristics of *Scedosporium* are the presence of spores in cultures, not commonly observed in other fungi, and the more irregular branching pattern of hyphae, but even those could not guarantee the diagnosis [23,24]. The microscopic morphology of colonies, including the presence of species-specific subhyaline, obovoidal, or ellipsoidal conidia and conidiogenous cells, aids presumptive identification [25]. Matrix-laser desorption/ionization mass spectrometry (MALDI-TOF/MS) has been evaluated as a fast and accurate identification method [26]. Although *Scedosporium* may appear susceptible in vitro to the applied drug according to the mycogram, this genus can demonstrate unpredictable resistance even to azoles, leading to recurrences of scedosporiosis [27]. Current recommendations based on the Global Guideline for the Diagnosis and Management of Rare Mold Infections support the use of voriconazole as a first-line treatment across all patterns of organ involvement. Combination treatment with voriconazole plus liposomal amphotericin B or amphotericin B lipid complex, echinocandins, or terbinafine can be used as first-line alternatives. Second-line treatment choices include isavuconazole, posaconazole, or itraconazole [25]. Better survival has been reported in patients suffering from cutaneous manifestations treated with voriconazole [28]. Spores being produced are phagocytosed by macrophages, while mononuclear and polymorphonuclear leukocytes damage the hyphae of *Scedosporium*, suggesting that interferon-γ or granulocyte-colony-stimulating factor (G-CSF) could be initiated as alternative treatments [29,30]. Herein, we report a case of recurrent localized scedosporiosis in a woman with RA receiving immunosuppressants treated with voriconazole and itraconazole. Further to this case, a literature review regarding *Scedosporium* infection in RA patients is being presented.

## 2. Case Report

An 81-year-old woman with a known history of RA since 1999, currently receiving methylprednisolone 6 mg daily, walked into our clinic in August 2022, presenting with swelling in her right hand and forearm. She reported being in her usual state of health until four days ago, when she developed a fever (Tmax 38.2 °C) and swelling of her right upper limb (Figure 1a). She resided in a rural area with no direct exposure to pets and no reported outdoor activities or recent travel.

For the last thirteen years, the patient was on regular treatment with tocilizumab (8 mg/kg q 4 weeks), more specifically from August 2009 until April 2022, when she was diagnosed with COVID-19, and was not able to restart tocilizumab since then. SARS-CoV-2 test was performed because some members of her family were infected with the COVID-19 virus, but she remained asymptomatic with no need for antiviral intervention or any modification of the steroid dose. For the last two months, she has been on a stable dose of methylprednisolone at 6 mg daily. Previously, she received methotrexate at a maximum dose of 15 mg weekly from 2009 until May 2017 and leflunomide 20 mg daily from 2005 until April 2009 for RA, which had been discontinued due to hepatotoxicity and secondary failure, respectively. Her medical history was significant for atrial fibrillation, mitral valve regurgitation, arterial hypertension, dyslipidemia, osteoporosis, and prediabetes. She was a non-smoker and a non-alcohol consumer.

On clinical examination, cellulitis of the right hand and forearm was noted. At the same time, a cystic nodule on the dorsal wrist surface, two visible micropustules, and two superficial ulcers surrounded by necrotic skin lesions were present. The patient reported that at the suffering region, an intravenous (IV) catheter had been placed approximately one month ago during her previous hospitalization.

A musculoskeletal ultrasound revealed a cystic nodule with circumferential Power Doppler present, indicative of inflammation, and severe tenosynovitis of the extensors of the right wrist (Figure 1b). An ultrasound-guided aspiration of 2 mL of purulent fluid was performed and sent for microscopy and culture. Her laboratory exams were remarkable for elevated erythrocyte sedimentation rate (46 mm/h) and C-reactive protein (68 mg/L; normal range < 6 mg/L). The white blood cell count was 12,540/μL, with neutrophils and lymphocytes being 10,500/μL and 1240/μL, respectively.

The patient was initiated on empirical antibiotic treatment with piperacillin/tazobactam and teicoplanin and remained afebrile. On the fourth day of hospitalization, we were informed that the culture of the purulent fluid tested positive for non-Candida fungus, so itraconazole was added to the antibiotics to include possible sporotrichosis. In the following days, a couple of new cystic lesions were observed. On the 12th day of hospitalization, the fungal isolate was identified as *Scedosporium* spp. The fungal blood cultures tested negative. Therefore, itraconazole was discontinued, and the patient was started on IV voriconazole (300 mg bid) for day 1, continuing with 200 mg bid. Meanwhile, computed tomography of the right upper hand was performed to exclude osteomyelitis, as the patient could not perform magnetic resonance imaging due to ferromagnetic foreign bodies. The patient received IV voriconazole for 14 days, then switched to per os (PO) voriconazole 200 mg bid due to adverse events, especially elevated liver function tests (AST 87 IU/L, ALT 85 IU/L, γGT 1383 IU/L, normal < 35 IU/L), and after ruling out other causes (i.e. drug reaction, infection and ischemia) with appropriate laboratory and imaging tests. The patient was already improving clinically at the time of this incident, and therefore, a decision to switch to per os voriconazole could have allowed her to continue the medication at home, anticipating a potential beneficial effect on her liver chemistry. One week after the initiation of voriconazole, no new skin nodules appeared, and gradual clinical improvement was noticed (Figure 1c).

Using MALDI-TOF/MS, *Scedosporium apiospermum* was identified, and the antifungal susceptibility testing determined susceptibility to voriconazole (MIC 0.25 mg/L), itraconazole (MIC 1 mg/L), and posaconazole (MIC 1 mg/L) via microdilution test. Due to persistently elevated transaminases, PO voriconazole was reduced to 200 mg daily after seven days of initiation for one month before being discontinued. The patient’s upper limb appeared to have neither skin lesions nor edema. During this period, the patient suffered some minor flares of RA, but they were handled with transient increases in steroid dose. No immunomodulatory or immunosuppressant agent was introduced.

On February 2023, four months after voriconazole was stopped, the patient was admitted to our clinic with a fever (Tmax 38.9 °C) and fatigue. No other sign was significant at clinical evaluation, except for a nodule on the dorsal surface of the wrist, which was aspirated, unveiling pus that was sent for microbiologic examination (Figure 2). Her laboratory exams were noteworthy for elevated CRP (78 mg/L) and lymphopenia (absolute number 450/μL, normal > 1000μL) with a normal number of white blood cells. The purulent fluid from the hand cystic nodule tested positive for *Scedosporium* spp., and itraconazole 100 mg bid was added to the drug regimen. Itraconazole was preferred for the recurrence of *Scedosporium* over voriconazole because the latter had only caused moderate adverse events in our patient. The patient responded well to treatment, with no new skin lesions and a decrease in the size of the existing lesions. After 11 days of PO itraconazole, she was discharged with close follow-up and a plan to continue itraconazole for an extended period of time, depending on her clinical response and routine blood tests, as itraconazole level measurement was unavailable. The steroid dose remained the same as at the patient’s admission.

## 3. Materials and Methods

A search in PubMed and Scopus was performed on 28 April 2023, using the keywords “scedosporium” OR “pseudallescheria” OR “lomentospora” AND “rheumatoid arthritis”, putting no time or language limitations on publications. Nine articles showed up; one out of those was excluded because it referred to pathogenetic mechanisms, and the second one was inaccessible. Three more eligible articles have been identified through the references of other reviews, but one of them could not be retrieved (Cremer, G.; Bournerias, I.; Mhalla, S. et al.—Scédosporiose cutanée non mycétomateuse chez un patient immunodéprimé, J. Mycol. méd., 4:111-114, 1994). The final search ended up with a total of nine articles. All articles, including titles, abstracts, and full texts, were independently reviewed by two authors (A.I.V. and P.V.V.). The method of this systematic search is shown in a block diagram (Figure 3). Parameters regarding clinical manifestations and treatment outcomes were recorded (Table 1). For the data analysis, IBM SPSS Statistics V25 has been used.

## 4. Results

In the literature search, nine cases of scedosporiosis in patients with RA were identified. Demographics, baseline drug record, *Scedosporium* spp., organ involvement, treatment, and outcome have been recorded for all patients, including the present case. The median age of patients was 68.5 years old (SD ± 9.7), with a female predominance of approximately 2:1. All patients were on chronic steroid treatment. Almost half of them (40%) were on monotherapy with steroids, whereas one was on concomitant treatment with methotrexate, one with azathioprine, two with calcineurin inhibitors, and one with bucillamine. Only one patient was on multiple immunomodulatory and immunosuppressive drugs, receiving methotrexate, hydroxychloroquine, and etanercept for rheumatoid arthritis. The majority of patients (60%) were treated with prednisone equivalent to more than 7.5 mg daily. The evolution of symptoms ranged from three weeks to 12 months. Most RA cases described are related to *Scedosporium apiospermum* (60%), followed by *Pseudallescheria boydii* (*Scedosporium boydii*) (20%), *Scedosporium dehoogii*(10%), and *Lomentospora prolificans* (10%), frequently affecting the skin (60%), causing osteomyelitis (30%), followed by lung, central nervous system, and eye manifestations (10% each). For those suffering from cutaneous scedosporiosis, the upper limbs were twice as often affected as the lower limbs (four patients with upper limb scedosporiosis vs. two with lower limb). Moreover, 57.1% of patients reported previous minor or major trauma to the affected area. The typical clinical course of skin lesions described includes initially a nodule at the site of a minor injury, followed by multiple erythematous, ulcerated, or suppurative nodules in the presence of micropustules, which could be slightly painful or painless, progressing, if left untreated, to subcutaneous swelling. At times, the lesions can also be pruritic. Regarding the antifungal treatment, itraconazole was administered to 40%, voriconazole to 30%, and one patient received an undefined antifungal treatment. A mycogram was performed in five cases (50%) with similar sensitivities to the azoles tested (voriconazole, itraconazole, miconazole, ketoconazole, and posaconazole). The median treatment duration was 10 months. There were two patients (20%) who switched to another azole because of adverse events related to treatment. Five patients out of ten were exposed to surgical procedures. Of the nine reported cases in the literature, recurrences were noted in three of them, while there was a patient with a brain abscess who passed away. Of those who relapsed, no more than one scedosporiosis recurrence was observed. Concerning the immunomodulatory treatment for RA, there is a lack of available information regarding steroid dosing and possible re-initiation of DMARDs in the vast majority of cases.

## 5. Discussion

We described the case of a female patient with RA, previously treated with tocilizumab and on chronic steroid use, who was eventually diagnosed with recurrent cutaneous scedosporiosis. Since the patient had a long-standing RA diagnosis on different immunosuppressants, it is well understood that her immune response to pathogens has been disrupted [40]. She was on treatment with tocilizumab, a monoclonal antibody approved in 2010 for RA that binds to the IL-6 receptor (IL-6R) and blocks IL-6 signaling [41]. In vitro studies suggest that tocilizumab impedes the recruitment of mononuclear cells, subdues neutrophil eradication, and inhibits T-cell activation [42,43,44]. IL-6 blockade implies immunomodulatory effects on the transition from innate to adaptive immunity. Several studies have reported a higher risk of IL-6 inhibitors, mostly tocilizumab, for serious infections, notably skin and soft tissue infections, compared with other biologic agents [45,46,47,48,49]. Nevertheless, the real impact of tocilizumab use on fungal infections is not well known.

Mice lacking the IL-6 response display defective immunity against fungi [50]. Studies have shown that *Scedosporium* hyphae are associated with the elevation of IL-6, which in turn promotes macrophage differentiation and B- and T-cell activation. This process is mandatory as hyphal structures are large enough not to be phagocytosed; instead, they are being attacked extracellularly by B- and T-cells [51]. CD4+ T-cells activated by fungal products promote the long-term migration and activation of monocytes and macrophages, preventing invasive fungal infections [41].

In line with a tocilizumab post-marketing all-case surveillance study, the incidence rate of cryptococcosis in tocilizumab-treated patients with RA was 0.02% [52]. One should keep in mind that studies are usually conducted in non-endemic areas for such fungal infections. Different studies are usually focused on different types of infections and measure specific outcomes, leading to a misconception of real-life risk [5]. Various tocilizumab dose regimens suggest a dose-dependent effect on infections, as the higher the doses of the drug or the shorter the administration intervals, the greater the risk for infections [53]. The likelihood of serious infections was roughly twice as high among those treated with TCZ 8 mg as opposed to 4 mg plus methotrexate, compared with those receiving methotrexate monotherapy [54]. In a pooled analysis of over 4000 RA patients with a mean treatment duration of TCZ of 2.4 years, 11 cases of invasive fungal infections were described: six cases of invasive candidiasis, one case of *Pneumocystis jirovecii* pneumonia, and one of cryptococcal pneumonia. No opportunistic infections were reported in the 1555 patients in the placebo arms of these studies [53]. Scarce real-world data are available regarding opportunistic infections, like fungal infections, and the data derived from clinical trials describe a limited but important risk for those, even though they were not designed with a special interest in them [5,55]. Risk factors for the development of severe infections in patients receiving TCZ have been reported to be concurrent or medical history of respiratory disorders, prednisolone dose at baseline ≥5 mg/day, and age ≥65 years [56]. Our patient had two or more of these factors: she had a relatively recent COVID-19 infection, which can predispose to subsequent fungal infections; she was older; and she was receiving corticosteroids. Indeed, COVID-19 has been associated with the reactivation of latent viral infections or the development, a few days or weeks later, of bacterial or fungal diseases. The reason could be COVID-19-related persistent lymphopenia or immunosuppression related to other viral-induced mechanisms [57].

Scedosporiosis may affect people of all ages, with the currently described cases aged from 2 to 92 years old, and this fungus has a long incubation period before it causes illness [12]. Its differential diagnosis with *Aspergillus*, *Fusarium*, and other hyaline hyphomycetes is crucial, with therapeutic and prognostic implications. Unfortunately, it is common for *Scedosporium* to be misidentified, and that could be a reason for its relatively low incidence. Two characteristics that differentiate *Scedosporium* spp. from other fungi are the production of ovoid conidia and the hyphal irregular branching pattern [58]. This genus responds well to triazoles, contrary to traditional antifungals, with voriconazole being proposed as the first-choice drug. There is no optimal dose or duration of treatment for scedosporiosis because of its rarity. Moreover, one should consider that antifungal sensitivities do not always correspond to in vivo reality and that azoles are fungistatic drugs, partially explaining the trend of *Scedosporium* to relapse [12,34]. However, voriconazole shows fungicidal activity against *Scedosporium apiospermum* in vitro [59]. On the other hand, *Scedosporium apiospermum* is resistant to amphotericin B [60]. For those with cutaneous scedosporiosis who could take advantage of surgical excision, a thorough debridement should be performed, allowing systemic antifungals to better penetrate soft tissues. In one of the largest studies on *Scedosporium spp*.treatment, it was reported that 57% of patients responded to administered drugs after a median time period of 103 days [28]. Recurrences seem to be unpredictable, regardless of susceptibility to azoles and duration of treatment, even though shorter courses may incur greater risk. In addition, it is assumed that *Scedosporium* spp. could colonize tissues, serving as a reservoir for recurrences [58,61].

Another point to consider is the voriconazole-induced hepatotoxicity in our patients. Some studies from Japan propose that being a carrier of a mutant CYP2C19 is related to poor metabolization of voriconazole, having excluded other drug interactions [62,63]. No major drug interaction was identified in our patient, and genetic testing was not available. In addition, dual antifungal regimens have been suggested in refractory cases of scedosporiosis after voriconazole failure, such as micafungin plus GM-CSF, aiming to boost phagocytosis and an oxidative burst leading to hyphal deterioration. A single case of an HIV-positive pediatric patient with scedosporiosis has been described, where salvage therapy with IFN-γ, G-CSF, and antifungals was applied [64]. Another dual treatment proposed is itraconazole plus terbinafine, which has also been administered in cases of adverse events attributed to voriconazole or in cases of resistant scedosporiosis [37]. The mortality of scedosporiosis is known to remain high, exceeding 72.7% in severely immunocompromised patients or disseminated cases [65].

The growing indications of steroid use, disease-modifying anti-rheumatic drugs (DMARDs), and biologic agents in patients with autoimmune rheumatic diseases, in conjunction with more sensitive infectious disease diagnostic techniques, may play a role in the increasing recognition of fungal infections. Thus far, there is no indication for pharmaceutical prophylaxis while on treatment with a biologic drug, but caution should be exercised for early diagnosis [5]. Physicians should be aware that tocilizumab could restrain early inflammatory symptoms and the elevation of CRP, delaying proper diagnostic evaluation and leading to fatal outcomes. Immunomodulation by blocking the IL-6 pathway in fungal infections could prove a double-edged sword, providing a state of immune tolerance at the expense of fungal persistence in the host.

We concluded that our patient was diagnosed with scedosporiosis quite early and received the proper treatment, avoiding fungal dissemination in this immunosuppressed patient. Due to treatment-associated adverse events, she did not receive an extended course of voriconazole (60 days or more), possibly preventing *Scedosporium* from being eliminated. As a result, and due to the chronic acquired immunodeficiency, scedosporiosis recurred. Whole genome sequencing or other molecular methods to compare the two strains were not available at our institution. Antifungal susceptibilities had some differences between the two strains; however, treatment with azoles had proceeded. Our patient did not report subsequent exposure to environmental fungi. She lived in an apartment, and due to her frailty, multiple comorbidities, old age, and hospitalizations, she had limited contacts and activity. She was cared for by her daughter, who reported that she followed all measures of hygiene that were recommended for an immunocompromised person. We have considered that the infection relapsed because the patient had not completed the maximum recommended duration of antifungal treatment, the infection occurred in proximity to the initial lesions at the upper extremities, and she continued to have risk factors for recurrence, like lymphopenia and corticosteroid use. However, an infection with more than one strain of *Scedosporium* species could not be ruled out.

The review of the literature revealed only three cases treated with TCZ who had a fungal infection: one suffering from RA diagnosed with *Fusarium* endopthalmitis, another suffering from RA diagnosed with allergic bronchopulmonary aspergillosis, and the last one, a patient suffering from Castleman’s disease diagnosed with disseminated cryptococcosis [52,66,67]. In the first case, TCZ was continued, with a potentially larger risk for fungal eye infections, albeit the treatment was successful, possibly due to the limited affected area. In the second case, TCZ was permanently discontinued and replaced by a calcineurin inhibitor. In the last case, TCZ was temporarily discontinued and re-initiated after three months of azole discontinuation. The few reported cases could be an indicator of the relative safety of TCZ in that type of infection, irrespective of the role of the IL-6 signaling pathway in fungal infections, implying the involvement of alternative pathogenetic pathways. The physician should weigh the benefits and harms, depending on the affected organs and the dominant clinical problem between the fungal infection and RA. To our knowledge, no other case of scedosporiosis in a patient with RA treated with TCZ and steroids has been described. Evidently, though, since the data is so limited, no definite associations can be made.

Questions that remain to be answered in the future are the optimal dose of antifungal agents and the proper duration of their administration, considering the potential drug-related adverse events and the real risk of recurrence, as well as the safety of the re-introduction of immunosuppressive treatment or even the concurrent use of prophylactic antifungal agents. Towards this direction, a Working Group on Pseudallescheriasis/Scedosporium infections was founded in 2002, now named the “Joined ECMM/ISHAM Working Group”, whose aim is to obtain insight into the occurrence and genetic variability of these fungi and also provide data on possible sources of contamination and infection routes. A special focus has been given to immunocompromised patients, who are at higher risk of acquiring such infections.

In conclusion, clinical alertness regarding uncommon infections in patients with rheumatologic diseases on immunomodulatory agents, as well as early diagnosis of fungal infections, are essential for effective treatment.

## Figures and Tables

**Figure 1 jof-09-00683-f001:**
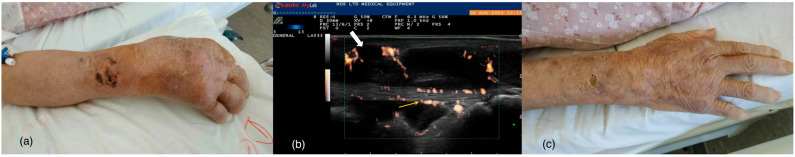
(**a**) Swelling of the patient’s right hand at clinical evaluation, (**b**) Musculoskeletal ultrasound demonstrating a cyst (white arrow) and severe tenosynovitis with present Power Doppler (yellow arrow), (**c**) Patient’s right hand at discharge.

**Figure 2 jof-09-00683-f002:**
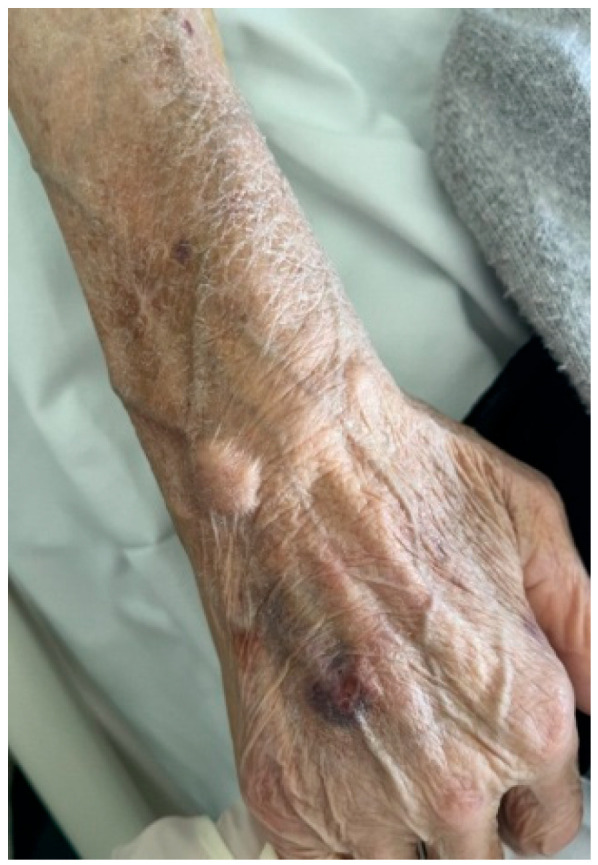
Recurrence of scedosporiosis on the patient’s right hand.

**Figure 3 jof-09-00683-f003:**
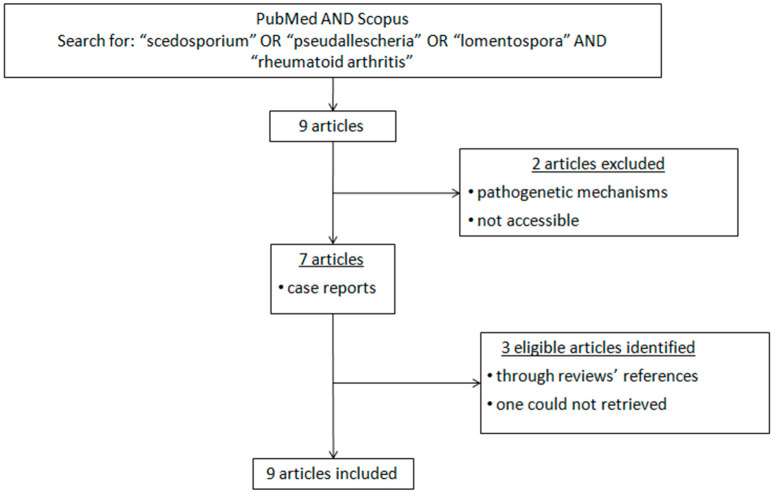
Methodology flowchart.

**Table 1 jof-09-00683-t001:** *Scedosporium* spp. infection in patients with rheumatoid arthritis.

Author, y	Age (y)/Sex	Underlying Disease	Therapy at Presentation	Pathogen	Organ Involvement	Treatment	Outcome
Murayama T. et al. (1998) [31]	69/F	RA, tuberculous pleurisy, and gold-induced interstitial pneumonia	MPz 4 mg/d, bucillamine	Scedosporium apiospermum	Lungs	MPz discontinuation	Stable; no recurrence reported
Khan S.A. (2000) [32]	Middle aged/F	RA	Pz; unknown dose	Pseudallescheria boydii	Central nervous system (brain abcess)	Surgical excision and IV antifungal agent (not determined)	Death
Fays S. et al. (2002) [33]	65/M	RA	MPz; unknown dose	Scedosporium apiospermum	Skin and bone	ITZ	Successfully treated; no recurrence reported
Chaveiro M.A. et al. (2003) [34]	63/F	RA and diabetes mellitus	MPz 10 mg/d and Cyclosporin 100 mg/d	Scedosporium apiospermum	Skin	ITZ 400 mg/d Cyclosporin	Almost total resolution at 4 weeks; no recurrence reported
Gottesman-Yekutieli T. et al. (2011) [35]	56/F	RA and hip replacement	Pz 20 mg/d MTX	Pseudallescheria boydii	Bone	Removal of the prosthetic joint; cement spacer impregnated with ITZ; oral VCZ 800 mg on Day1 → 400 mg/d	Recurrence 6 weeks later while on VCZ; hip reoperation VCZ for 10 months
Canet J.J. et al. (2011) [36]	69/F	RA, Sjoegren syndrome, arterial hypertension, chronic renal failure, and chronic hepatitis C	Pz 10 mg/d	Scedosporium apiospermum	Subcutaneous	Surgical drainage and ITZ 200 mg/d	Almost total resolution at 10 weeks; no recurrences reported
Sakata Y. et al. (2017) [37]	77/M	RA	Betamethasone 1.25 mg/d and Tacrolimus 3 mg/d	Scedosporium dehoogii	Skin	Oral VCZ 400 mg/d and local hyperthermia	Improvement, but VCZ- induced liver dysfunction; switch to ITZ 200 mg/d, but worsened; add on terbinafine 125 mg/d for 10 months (ITZ for 6 months)
Lee J. et al. (2020) [38]	68/M	RA	Pz 7.5 mg/d, HCQ 200 mg/d, and MTX 10 mg/wk Etanercept 50 mg/wk	Lomentospora prolificans	Skin and bone	Oral VCZ 250 mg BD and terbinafine 250 mg BD VCZ bone cement and surgical debridement	Improvement; no recurrence; VCZ and terbinafine for at least 12 months
Palma-Fernández R. et al. (2021) [39]	78/F	RA	Pz 10 mg/d and Azathioprine 100 mg/d	Scedosporium apiospermum	Eye	Surgical debridements; topical eye solution; systemic therapy with VCZ and steroids	Improvement; reccurence;–second therapeutic course for 12 months
Present case	81/F	RA, atrial fibrillation, mitral valve regurgitation, arterial hypertension, dyslipidemia, and osteoporosis	MPz 6 mg/d	Scedosporium apiospermum	Skin	IV VCZ 400 mg/d; Switch to PO VCZ 200 mg/d (due to VCZ-induced liver dysfunction) for 1 month	Reccurence; ITZ 200 mg/d

F: female; M: male; MTX: methotrexate; HCQ: hydroxychloroquine; y: year; d: day; MPz: methylprednisolone; Pz: prednisone; IV: intravenous; PO: per os; ITZ: itraconazole; VCZ: voriconazole; BD: twice daily.

## Data Availability

Not applicable.

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
