# Peer review of "Recurrent Scedosporium apiospermum Cutaneous Infection in a Patient with Rheumatoid Arthritis: The Potent Role of IL-6 Signaling Pathway Blockade: A Case-Based Review"

_jof, 2023, doi:10.3390/jof9060683_

Round 1

Reviewer 1 Report

The study authors report of an elderly lady with rheumatoid arthritis on immunosuppressive therapies who developed localized recurrent cutaneous Scedosporium infection. They further review the literature to report on 8 other cases of Scedopsorium infection occurring in RA patients on immunosuppressives.

Overall, the case report is well written and with a comprehensive introduction. Table 1 provides a good summary of this rare infection in RA patients. There are certain points as below that should be clarified prior to publication, some which would help increase the educative value of the manuscript.

Major Comments

1. The study authors report a diagnosis of COVID-19 in their patient 4 months prior to initial clinical presentation of cutaneous Scedosporium infection. COVID-19 is now known to be associated with later development of cutaneous fungal infections (Abhirami et al DOI: 10.4103/ijd.ijd_781_21). Additionally, the use of immunosuppressive medications for treatment of COVID-19 may also predispose to development of fungal and bacterial infections in the future. The authors should include details regarding the patients COVID-19 history and treatments she received for it. Additionally, they could consider elaborating in the Discussion about the potential role of COVID-19 in the pathogenesis of the  Scedosporium infection.

2. The study authors report that IV voriconazole was switched to oral voriconazole due to concern for elevated liver function tests. However, hepatic adverse effects are also possible with oral voriconazole and are not exclusive to the IV formulation. Was there a different reason for the switch from IV to oral? Did the patient have voriconazole drug levels tested? Was there workup for other potential causes of transient liver function test abnormalities performed? These are important details that need clarification, particularly as the patient was then reported to have received oral voriconazole for only one month.

3. The study authors reported on the antifungal susceptibilities of the Scedosporium isolate. It would be educative to the readers to elaborate on whether they employed CLSI or EUCAST guidelines for the susceptibility testing.

Minor Comments

1. The study authors should elaborate if there has been any history of outdoors activities (such as gardening), history of trauma, residence in urban/rural areas, pets exposure, recent travel or any other epidemiologic factors in her clinical history that may have played a role in her acquiring the fungal infection.

2. Page 3/14, line 113 – please elaborate on what is meant by “secondary inefficiency” in relation to leflunomide.

3. Page 3/14, Lines 122-123 – please elaborate on what is meant by “Power Doppler present”. It would also be helpful to indicate these with relevant arrows in the ultrasound photograph.

4. Page 3/14, Lines 125-126 – please include the full-forms of the abbreviations for ESR and CRP. Also please include the normal range for ESR (I believe it should be 0-20mm/hr) but there may be some readers who are not aware of it.

5. Page 3 of 14, lines 127-140 - The study authors should elaborate whether the patient had fungal blood cultures performed. This would be reasonable to do given her underlying immunocompromised status.

6. Figures 1 a,b,c – if feasible, larger sized images should be provided for better representation of their patient’s clinical findings.

7. Page 4 of 14, line 147 – the study authors report that the patient received “transient increases in steroid dose”. Please elaborate whether the patient was discharged on the same initial dose of methylprednisolone as when she was admitted.

8. Page 4 of 14, lines 153-154 – please include the normal laboratory parameters and normal units (cells/mm3) for the lymphocyte count.

9. Page 4 of 14, lines 159-160 – please mention if the patient had itraconazole levels tested as an outpatient and their test results.

10. Page 5 of 14, lines 201-202 – the authors should consider rephrasing the following sentence “The patients that suffered a recurrence compared to those who did not, were half and half” into something along the lines of “Of the eight reported cases in the literature, recurrences were noted in half of them”.

11. Table 1 – the study authors should expand on what antifungal agents were used for the patient case documented by Khan et al, if that data is available

12. In the Discussion, the study authors should consider adding an additional point related to resistance of Scedosporium apiospermum to Amphotericin B (a point that has been raised in the literature – for example, Ellis et al DOI: 10.1093/jac/49.suppl_1.7), to further create awareness about this for the readers.

Author Response

On behalf of our team, I would like to thank the reviewers for taking the time to evaluate our manuscript and make their valuable comments. Their suggestions are highly appreciated and were taken into thorough consideration for the improvement of our manuscript. We have revised our manuscript trying to address all the comments and formatting issues highlighted by the editorial team and the reviewers.

We believe that after these changes recommended by the reviewers, our manuscript has been further improved. You can find our point-by-point responses to the editorial team’s comments below.

Again, we are very thankful for taking into consideration our manuscript for publication in Journal of Fungi.

Reviewer#1

Major Comments

  • Comment 1: The study authors report a diagnosis of COVID-19 in their patient 4 months prior to initial clinical presentation of cutaneous Scedosporium infection. COVID-19 is now known to be associated with later development of cutaneous fungal infections (Abhirami et al DOI: 10.4103/ijd.ijd_781_21). Additionally, the use of immunosuppressive medications for treatment of COVID-19 may also predispose to development of fungal and bacterial infections in the future. The authors should include details regarding the patients COVID-19 history and treatments she received for it. Additionally, they could consider elaborating in the Discussion about the potential role of COVID-19 in the pathogenesis of the Scedosporium infection.

Reply: We would like to thank the reviewer for his/her insightful comment. Our patient was tested positive for COVID-19, without ever having any symptoms, so no intervention was needed, except for temporarily discontinuing the immunosuppressive medications she received. Corticosteroids were continued in the same dose. The test was performed, because some members of her family were infected with COVID-19 virus. Indeed, COVID-19 has been associated with the reactivation of latent viral infections or the development, few days or weeks later, of bacterial or fungal diseases. The reason could be persistent lymphopenia, immunosuppression related to other viral induced mechanisms or receipt of immunomodulatory treatment for COVID-19. (https://doi.org/10.1371/journal.pone.0271795). As stated in the text (page 6, line 260), “Risk factors for the development of severe infections in patients receiving tocilizumab have been reported to be concurrent or medical history of respiratory disorders, prednisolone dose at baseline ≥5 mg/day, and age ≥65 years (reference 47)”. Our patient had two or more of these factors, as she had a relatively recent COVID-19 infection, which can predispose to subsequent fungal infections, she was older and was receiving corticosteroids.

  • Comment 2: The study authors report that IV voriconazole was switched to oral voriconazole due to concern for elevated liver function tests. However, hepatic adverse effects are also possible with oral voriconazole and are not exclusive to the IV formulation. Was there a different reason for the switch from IV to oral? Did the patient have voriconazole drug levels tested? Was there workup for other potential causes of transient liver function test abnormalities performed? These are important details that need clarification, particularly as the patient was then reported to have received oral voriconazole for only one month.

Reply: We would like to thank the reviewer for this very important comment. When our patient presented with elevated liver enzymes, other causes (i.e. drug reaction, infection, ischemia) were ruled out with appropriate laboratory and imaging tests. Unfortunately, therapeutic drug monitoring of voriconazole is not available at our institution. Our patient was already improving clinically at the time of this incident, and therefore, a decision to switch to per os voriconazole, could have allowed her to continue the medication at home, anticipating also, a potential beneficial effect on her liver chemistry.

  • Comment 3: The study authors reported on the antifungal susceptibilities of the Scedosporium It would be educative to the readers to elaborate on whether they employed CLSI or EUCAST guidelines for the susceptibility testing.

Reply: We thank the reviewer for highlighting the need for this clarification. EUCAST guidelines were used for the susceptibility testing.

Minor Comments

  • Comment: The study authors should elaborate if there has been any history of outdoors activities (such as gardening), history of trauma, residence in urban/rural areas, pets exposure, recent travel or any other epidemiologic factors in her clinical history that may have played a role in her acquiring the fungal infection.

Reply: Reported in page 3, line 110). And already reported in page 3, line 127 that at the suffering region, an intravenous (IV) catheter was previously placed.

  • Comment: Page 3/14, line 113 – please elaborate on what is meant by “secondary inefficiency” in relation to leflunomide.

Reply: Corrected accordingly.

  • Comment: Page 3/14, Lines 122-123 – please elaborate on what is meant by “Power Doppler present”. It would also be helpful to indicate these with relevant arrows in the ultrasound photograph.

Reply: Amended accordingly.

  • Comment: Page 3/14, Lines 125-126 – please include the full-forms of the abbreviations for ESR and CRP. Also please include the normal range for ESR (I believe it should be 0-20mm/hr) but there may be some readers who are not aware of it.

Reply: Full- forms for ESR and CRP provided. Normal ESR normal levels are calculated by adding 10 to the patient age devided by 2.

  • Comment: Page 3 of 14, lines 127-140 - The study authors should elaborate whether the patient had fungal blood cultures performed. This would be reasonable to do given her underlying immunocompromised status.

Reply: Fungal blood cultures were negative- information added in page 3, lines 141-142.

  • Comment: Figures 1 a,b,c – if feasible, larger sized images should be provided for better representation of their patient’s clinical findings.

Reply: Images are provided in the largest possible size. Also explanatory arrows have been added.

  • Comment: Page 4 of 14, line 147 – the study authors report that the patient received “transient increases in steroid dose”. Please elaborate whether the patient was discharged on the same initial dose of methylprednisolone as when she was admitted.

Reply: Information provided in page 4, lines 172-173.

  • Comment: Page 4 of 14, lines 153-154 – please include the normal laboratory parameters and normal units (cells/mm3) for the lymphocyte count.

Reply: Normal ranges and units added as asked.

  • Comment: Page 4 of 14, lines 159-160 – please mention if the patient had itraconazole levels tested as an outpatient and their test results.

Reply: Itraconazole level monitoring was not available at our institution. Details added in page 4, line 172.

  • Comment: Page 5 of 14, lines 201-202 – the authors should consider rephrasing the following sentence “The patients that suffered a recurrence compared to those who did not, were half and half” into something along the lines of “Of the eight reported cases in the literature, recurrences were noted in half of them”.

Reply: Amended accordingly.

  • Comment: Table 1 – the study authors should expand on what antifungal agents were used for the patient case documented by Khan et al, if that data is available

Reply: It is not stated what kind of antifungal agents were used. Also added in Table 1.

  • Comment: In the Discussion, the study authors should consider adding an additional point related to resistance of Scedosporium apiospermum to Amphotericin B (a point that has been raised in the literature – for example, Ellis et al DOI: 10.1093/jac/49.suppl_1.7), to further create awareness about this for the readers.

Reply: The point related to resistance of Scedosporium apiospermum to Amphotericin B has been added to the discussion, page 6, line 282.

Reviewer 2 Report

The manuscript "jof-2284308" reviews a case of scedosporiosis complicating rheumatoid arthritis. Only a handful of these cases have been previously described and it's therefore useful to have this case report in the literature. However, several key details are lacking which should be added in order for this case review to be useful for future clinical implications.

- L48 Write out DMARDs

- L59-60 These species names have been updated to match phylogeny. See Kidd et al 2023 Open Forum Infec Dis.

- L87 in vitro should be italic

- L93 terbinafin should be terbinafine

- L104 currently, describe for how long tocilizumab has been taken

- L111-112 at the same time, what time period. Detail needed

- L112 "previously she received" when and for how long on what doses.

- L160 "Extended period" How long initially?

- L163: Lomentospora should be added here as it is now part of the same complex. Please perform, will not add many articles because this is rare but would be great for completion.

- L185-186 Update species names

- L248: ages are too specific, it can affect anyone really.

Author Response

On behalf of our team, I would like to thank the reviewers for taking the time to evaluate our manuscript and make their valuable comments. Their suggestions are highly appreciated and were taken into thorough consideration for the improvement of our manuscript. We have revised our manuscript trying to address all the comments and formatting issues highlighted by the editorial team and the reviewers.

We believe that after these changes recommended by the reviewers, our manuscript has been further improved. You can find our point-by-point responses to the editorial team’s comments below.

Again, we are very thankful for taking into consideration our manuscript for publication in Journal of Fungi.

Reviewer#2

  • Comment: L48 Write out DMARDs

Reply: Amended accordingly.

  • Comment: L59-60 These species names have been updated to match phylogeny. See Kidd et al 2023 Open Forum Infec Dis.

Reply: Species names have been updated.

  • Comment: L87 in vitro should be italic

Reply: Amended accordingly.

  • Comment: L93 terbinafin should be terbinafine

Reply: Amended accordingly.

  • Comment: L104 currently, describe for how long tocilizumab has been taken

Reply: Already described in page 3, lines 112, but also added the exact timeline (line 113).

  • Comment: L111-112 at the same time, what time period. Detail needed.

Reply: Details added.

  • Comment: L112 "previously she received" when and for how long on what doses.

Reply: Amended accordingly.

  • Comment: L160 "Extended period" How long initially?

Reply: The time plan was indefinite and would be determined by the patient’s clinical response and the monitoring of routine lab tests.

  • Comment: L163: Lomentospora should be added here as it is now part of the same complex. Please perform, will not add many articles because this is rare but would be great for completion.

Reply: One article was added (reference 64) and the results were modified accordingly.

  • Comment: L185-186 Update species names

Reply: Species names have been updated.

  • Comment: L248: ages are too specific, it can affect anyone really.

Reply: A comment has been added in page 6, line 269.

Reviewer 3 Report

The case is interesting because of the limited number of patients reported with these associated diseases. Having said that, I think that a discussion/review of the literature using only 8 published manuscripts is too early, and any possible conclusion or speculation is too premature and likely to be modified as a significant number of cases accumulates. So, I think this journal is not the best place to publish this case, a more specialized journal in the clinical setting or in case reports seems to be suitable.

Related to the case are some questions the manuscript does not address and are relevant to improve the presentation.

Does the patient suffer from a chronic disease that could affect immunological fitness? Does she have diabetes mellitus? Did the white blood cells analysis show any significant disturbance when admitted the first time.? The fungus was isolated twice from the patient, and the way the manuscript is written gives the impression that is the same strain, which would make sense, but the authors should provide evidence that the two isolates have the same molecular markers to conclude indeed is the same isolated that was not fully eliminated from the patient. Another alternative to explore is that microbiological resolution of the infection occurred but the patient suffered reinfection from an external source of the organisms, such as her household.

Author Response

On behalf of our team, I would like to thank the reviewers for taking the time to evaluate our manuscript and make their valuable comments. Their suggestions are highly appreciated and were taken into thorough consideration for the improvement of our manuscript. We have revised our manuscript trying to address all the comments and formatting issues highlighted by the editorial team and the reviewers.

We believe that after these changes recommended by the reviewers, our manuscript has been further improved. You can find our point-by-point responses to the editorial team’s comments below.

Again, we are very thankful for taking into consideration our manuscript for publication in Journal of Fungi.

Reviewer#3

  • Comment: Does the patient suffer from a chronic disease that could affect immunological fitness?

Reply: Yes, she does. Please check page 1, lines 30-38.

  • Comment: Does she have diabetes mellitus?

Reply: At that moment she was diagnosed with prediabetes. Information added in page 3, line 123.

  • Comment: Did the white blood cells analysis show any significant disturbance when admitted the first time?

Reply: She had no significant disturbance in blood cell analysis. Exact numbers provided in page 3, lines 134-135.

  • Comment: The fungus was isolated twice from the patient, and the way the manuscript is written gives the impression that is the same strain, which would make sense, but the authors should provide evidence that the two isolates have the same molecular markers to conclude indeed is the same isolated that was not fully eliminated from the patient. Another alternative to explore is that microbiological resolution of the infection occurred but the patient suffered reinfection from an external source of the organisms, such as her household.

Reply: We would like to thank the reviewer for this important point. Unfortunately, whole genome sequencing, or other molecular methods to compare the two strains were not available at our institution. Antifungal susceptibilities had some differences between the two strains, however treatment with azoles had preceded. Our patient did not report subsequent exposure to environmental fungi. She lived in an apartment and due to her frailty, multiple comorbidities, old age and hospitalizations, she had limited contacts and activity. She was cared by her daughter and reported that she kept all measures of hygiene that were recommended for an immunocompromised person.

We have assumed that the infection relapsed because the patient had not completed the maximun recommended duration (60 days are recommended) of antifungal treatment, the infection occurred in proximity to the initial lesions at the upper extremities and she continued to have risk factors for recurrence, like lymphopenia and corticosteroid use. However, an infection with more than one strain of Scedosporium species cannot be ruled out.

Round 2

Reviewer 3 Report

Most of my previous concerns were addressed, thank you. The following point was not:  

The case is interesting because of the limited number of patients reported with these associated diseases. Having said that, I think that a discussion/review of the literature using only 8 published manuscripts is too early, and any possible conclusion or speculation is too premature and likely to be modified as a significant number of cases accumulates. So, I think this journal is not the best place to publish this case, a more specialized journal in the clinical setting or in case reports seems to be suitable.

The issue related to the two isolates' phylogenetic relationship remains to be addressed. I understand the institution does not have facilities to perform molecular analyses, but this is not a scientific reason for not including this essential data, I am sure several groups would be interested in collaborating with the authors. Regrettably, my first recommendation still stands.

Author Response

We would like to thank the reviewer for his/her insightful comments. Due to the rarity of patient cohorts reporting opportunistic fungal infections in patients with rheumatoid arthritis or other diseases, who are receiving anti-IL-6 agents, we have decided to share our experience, in view of previously published evidence. The rate of these infections, we believe, will not significantly increase over the next few years and thus this review can be useful and timely for colleagues in the field. In this regard, our aim was to raise clinical alertness regarding a potential correlation between IL-6 blockade and opportunistic fungal infections in patients with rheumatologic diseases. Evidently, though, since data is so far limited, no formal associations can be made. We have added this latter comment to the text (line 345-346).

At the time of study, unfortunately we had not sent the strains for molecular analysis to determine their phylogenetic similarity. This would have undoubtedly increased the scientific rigor and validity of our results. Future collaboration with centers performing WGS of bacterial and fungal strains, in Greece or abroad, is within our goals, for diagnostic, therapeutic and infection control practices.